# Zoonotic RVA: State of the Art and Distribution in the Animal World

**DOI:** 10.3390/v14112554

**Published:** 2022-11-18

**Authors:** Ricardo Gabriel Díaz Alarcón, Domingo Javier Liotta, Samuel Miño

**Affiliations:** 1Laboratory of Applied Molecular Biology (LaBiMAp), Faculty of Exacts, Chemical and Natural Sciences, National University of Misiones (UNaM), Posadas 3300, Misiones, Argentina; 2National Institute of Tropical Medicine (INMeT)—ANLIS “Dr. Carlos Malbrán”, Puerto Iguazú 3370, Misiones, Argentina; 3National Institute of Agricultural Technology (INTA), EEA Cerro Azul, National Route 14, Km 836, Cerro Azul 3313, Misiones, Argentina

**Keywords:** rotavirus A, constellations, genotype, exotic host, zoonotic

## Abstract

Rotavirus species A (RVA) is a pathogen mainly affecting children under five years old and young animals. The infection produces acute diarrhea in its hosts and, in intensively reared livestock animals, can cause severe economic losses. In this study, we analyzed all RVA genomic constellations described in animal hosts. This review included animal RVA strains in humans. We compiled detection methods, hosts, genotypes and complete genomes. RVA was described in 86 animal species, with 52% (45/86) described by serology, microscopy or the hybridization method; however, strain sequences were not described. All of these reports were carried out between 1980 and 1990. In 48% (41/86) of them, 9251 strain sequences were reported, with 28% being porcine, 27% bovine, 12% equine and 33% from several other animal species. Genomic constellations were performed in 80% (32/40) of hosts. Typical constellation patterns were observed in groups such as birds, domestic animals and artiodactyls. The analysis of the constellations showed RVA’s capacity to infect a broad range of species, because there are RVA genotypes (even entire constellations) from animal species which were described in other studies. This suggests that this virus could generate highly virulent variants through gene reassortments and that these strains could be transmitted to humans as a zoonotic disease, making future surveillance necessary for the prevention of future outbreaks.

## 1. Introduction

Rotaviruses (RVs) are among the most important gastrointestinal pathogens causing severe acute diarrhea in children, young mammals and birds worldwide [1]. In humans, gastroenteritis not professionally managed may lead to death. Rotavirus A (RVA) belongs to the Reoviridae family (genus Rotavirus, Rotavirus species A). Annually, RVA is responsible for almost 500,000 deaths in infants under 5 years old, mainly in developing countries [2,3]. In animals, rotavirus infection can lead to colossal financial losses due to a decrease in productivity in livestock [4].

RVA has a segmented double-stranded RNA genome which consists of 11 segments enclosed in a triple-layered icosahedral capsid, encoding six viral structural proteins (VP1 to VP4, VP6 and VP7) and six nonstructural proteins (NSP1 to NSP6) [5]. Each genome segment, with the exception of gene 11 which encodes two proteins (NSP5 and NSP6), is monocistronic. The inner layer of the rotavirus virion is mainly composed of VP2, which encases VP1, the viral RNA-dependent RNA polymerase, and VP3, the viral capping enzyme [5]. The middle layer of the virion comprises VP6 trimers; VP7 and spikes of VP4 compose the outer layer. All nonstructural proteins, with the exception of NSP4, interact with nucleic acids and are involved in viral replication processes; NSP4 is responsible for morphogenesis [5].

In this review, the nomenclature based in the notation Gx-P[x]-Ix-Rx-Cx-Mx-Ax-Nx-Tx-Ex-Hx employed for the VP7-VP4-VP6-VP1-VP2-VP3-NSP1-NSP2-NSP3-NSP4-NSP5 encoding genes was considered, which compares complete rotavirus genomes [6].

Different rotavirus species are all grouped into the genus Rotavirus, including nine species (or groups) designated as RVA–RVD and RVF–RVJ, officially recognized by the International Committee on Taxonomy of Viruses (ICTV) [7]. Other proposed species designated as RVK and RVL [8] were considered, primarily due to partial genome sequences available so far [7]. In this study, we focused on RVA (Rotavirus species A), which is the most common and widely studied viral group because of its high prevalence worldwide in both mammals (including humans) and birds.

The main drivers of rotavirus diversity appear to be point mutations that occur continuously based on the high error rate of the RV polymerase and genome reassortments occurring by strain coinfections in hosts, often involving zoonotic transmission [1].

### 1.1. Economic Impact and Losses

Meat and dairy are important sources of nutrition for people and animals and big industries worldwide, hence the importance of their care. Global demand for both industries is growing: over the past 50 years, meat and dairy production has more than tripled [9]. The United States, Brazil and China, followed by Argentina, Australia and India, are the biggest productors of meat and milk around the world. Global production of poultry meat has increased rapidly over the last 50 years, growing more than 12-fold between 1961 and 2014. Similar to cattle production, the United States is the world’s largest producer, supplying more than 20 million tons of product in 2014. China and Brazil are also large poultry producers at 18 and 13 million tons, respectively. Pig meat production was around 112 million tons in 2014. China dominates global output, producing just short of half of total pig meat, followed by the United States, Germany, Spain and Brazil [9].

The average annual loss estimated for deaths caused by RVA infections in all these industries is counted in the millions. Moreover, the long-term effects of neonatal diarrhea on the health and performance of survivors could constitute an even greater loss [10,11].

### 1.2. One Health Approach

The “One Health” concept was introduced in the early 2000s. This concept promotes that human and animal health are interdependent and bound to the health of the ecosystems in which they exist [12]. Diseases of animal origin that can be transmitted to humans pose a global public health risk. It was estimated that 60% of existing human infectious diseases are zoonotic, and 75% of emerging infectious diseases in humans (including SARS-CoV-2, Ebola, HIV, Influenza virus, and Rotavirus) have an animal origin [13].

Analyzing the animal RVA genomic constellations will assist in better understanding of the evolution and ecology of RVA, and will contribute to the formulation of better vaccines by the selection of more adequate strains that evoke stronger and wider cross immunity, and better matching the co-circulating RVA strains. Then, preventing the disease in animals could prevent epidemic outbreaks in humans [14].

### 1.3. Animal’s RVA Genome in Human Strains

Since 2008, a new classification system of RVA based on the 11 genome segments has been used [15]. However, the dual nomenclature based on the two outer capsid proteins VP7 and VP4 that determine G and P genotypes, respectively, are still current [16]. To date, 42 G and 58 P types were described for RVA (https://rega.kuleuven.be/cev/viralmetagenomics/-virus-classification/rcwg) (accessed on 15 March 2022).

Animal rotaviruses can infect humans and cause disease through the ingestion of contaminated water. This is corroborated by the identification in humans of unusual rotavirus strains, with properties more commonly found in animals. These unusual human rotavirus strains may have arisen either as whole virions or as genetic reassortment between human and animal strains during co-infection of a single cell [17,18].

First, studies of specific rotavirus strains were associated with specific animal species; however, after the implementation of the new classification system, the host species’ descriptions of P and G types were improved. Human RVA strains that possess genes commonly found in animal rotaviruses have been isolated from infected children in both developed and developing countries. Strains such as G3 (found commonly in species such as cats, dogs, monkeys, pigs, mice, rabbits and horses), G5 (pigs and horses), G6 and G8 (cattle), G9 (pigs and lambs) and G10 (cattle) have been isolated from the human population throughout the world [19,20,21]. However, compared with human RVs, studies on animal RVs are scattered and limited, and to date, there are a dearth of global data on RV-related deaths in different animals [21]. Compiled information exists on the biology of rotavirus, structure and genome, proteins, classification, sero-groups immune response, diagnostic and other basic characteristics of this viruses [18]; however, this only covers a few animal species. The aim of this study was to compile RVA genetic information and genomic constellations, as well as hosts in which RVA have been described, to overview the current status and assess the distribution of RVA in the animal world.

## 2. Materials and Methods

The data were collected from the Virus Variation Rotavirus database (https://www.ncbi.nlm.nih.gov/genomes/VirusVariation/) (accessed on 15 March 2022), using the default settings. Briefly, the sequence search was carried out as “type” nucleotide, “species” rotavirus A, “region/country” any, “segment” any, “isolations source” any. Results were sorted by genotype and all typeable genotypes were used for analysis. Sequences without a genotype written in the database were analyzed by nucleotide BLAST (https://blast.ncbi.nlm.nih.gov/Blast.cgi) (accessed on 15 March 2022). Results were sorted by percentage of identity and searched for a sequence with a characterized genotype. Finally, we compared the results with reviews of RVA genotypes or reports of RVA in new hosts.

## 3. Results

We found reports of RVA in 82 different hosts (Table A1). Several animal RVA were detected by serological tests, electron microscopy or electropherotype (Table A1). Many deer-like artiodactyls, monkeys, kangaroo and grizzly bears were reported using at least one of these methods [22,23,24]. However, genetic analyses were performed on almost half of the hosts reported (40 host species), and there is information about their genotypes (9251 genotypes).

In this work, only animal RVs were analyzed in the following order: (1) the genomic constellations, (2) the GP[] combinations described and (3) the characteristics of particular genes. Hosts were analyzed individually or in groups, according to their relative importance. Although RVA strains with certain GP[] combinations are commonly found in particular hosts, whereas others are considered unusual for the same host, it should be noted that these observations are based on limited studies; therefore, they might eventually generate a bias due to the disproportionate relative numbers obtained from the database [19,20,21]. Approximately 110,000 human RVA strains were genotyped from 1996 to 2008, whereas only 1100 porcine and 3200 bovine RVs have been genotyped over the last three decades [19]. In our findings, the number of animal RVA sequences stored in NCBI had significantly increased by March 2021, reaching a total of 9251 reports. More than 2600 sequences were found in swine, and the same for cattle, plus 1100 in horse RVA strains (Table A2).

Hosts and constellations

At least one genomic constellation was reported in 34 out of 40 hosts. The hosts with more reported genomic constellations were pigs (43), cows (41), bats (23), horses (10) and monkeys (6). The genomic constellation structures of DS-1 (cows) and Wa (pigs) strains were found to be predominant (Table A3 and Table A4 respectively). Moreover, 22 genomic constellations were found in avian hosts such as the velvet scoter (duck), chicken, crow, gull, dove and guinea fowl (Table A7). In addition, 10 genomic constellations were found in single reports (bear, shrew, racoon, pheasant, gull, giraffe, red fox, sable antelope, crow, velvet scoter and vicuna). For other hosts, just one or two genes were reported, and no constellations were defined; these animals are the anatid (duck), donkey, clam and red squirrel.

Common GP[] genotype combinations

There are common combinations of VP7 and VP4 protein (GP[x]) such as G8P[1] that were found in dogs, monkeys, llamas, horses, goats, cows and guanacos, or G8P[14] found in alpacas, vicunas, llamas, sheep, guanacos and deer. Additionally, G3P[3] was present in hosts with no evolutionary relationship, such as dogs, monkeys, bats and horses. The most reported genotypes combinations are between G1 and G18 combined with P[1] to P[19], except for bats, which showed genotypes distributed between G20 and G36 and from P[6] to P[51]. Interestingly, birds possess different RVA gene genotypes genetically divergent from mammals and have their own gene frequencies and constellations [21,25,26] (Appendix A).

Prevalent VP7, VP4 and VP6 genotypes

Of all of the VP7 genes genotyped, G3 and G6 were the most reported ones (548 and 561, respectively). It was reported that the G3 genotype has the broadest host range [21,27], found commonly in species such as cats, dogs, monkeys, pigs, mice, rabbits and horses [18]. In accordance with these authors, we found that the G3 genotype is present in at least 18 different hosts, including humans, monkeys, rabbits, pigs, birds, cats, dogs, horses, mice, cows, goats, lambs, alpacas, bats, clams, llamas, rabbits and red squirrels [27,28,29,30]. On the other hand, the G6 genotype was present mainly in cows (531 reports). Similarly, G10, G14 and G23 were reported on many times, but were concentrated in just one host (cows, horses and pigs, respectively) (Appendix A).

In VP4 genes, the P[11] genotype has 249 reports (242 belong to cows), making it the most reported P genotype. On the other hand, the P[14] genotype was reported in 13 different hosts, 10 of which belong to the artiodactyl family. Most hosts have a single P-genotype or just a few ones (Appendix A).

In VP6 genes, I2 genotypes were found in 19 different hosts, with 254 reports. However, the most prevalent genotype was I5, with 289 overall reports (234 were in pigs) (Appendix A). 

## 4. Discussion

Cows

We found 41 genomic constellations reported in cows: 32 complete and 9 partial ones (Table A3). The DS-1-like genomic constellation G6-P[5]-I2-R2-C2-A3-N2-T6-E2-H3 is the most prevalent in cattle. The genotypes N2-T6-E2-H3 are constant in cows’ genomic constellations, and are present in all reports, except for 14 uncommon strains reported in Argentina, Japan, Uganda and South Korea (Table A3) [31,32,33].

In total, 19 VP7-VP4 genotype combinations were described in cattle (Appendix A). The most prevalent combinations were G6P[5], G6P[11] and G10P[11] that together represent the 40% of the reports. The remaining combinations were detected in fewer than 2% of cases, as previously reported [19].

Analyzing the VP7 genotypes, the G6, G8, G10 and G15 genotypes are mostly combined with the typical bovine DS-1-like genomic constellation. The most common worldwide bovine genotype is G6, followed by G10 in Europe, Asia and the Americas, and G8 in Africa and India (Appendix A). Regarding VP4, the most reported genotypes were P[11], P[5] and P[1] in accordance with previous studies [19] (Appendix A). The VP6 gene mainly reported in cows is the I2 genotype (73.5%), followed by I5 (25.5%) (Appendix A).

Several reports describing Wa-like genomic constellations were also found (Table A3. Porcine VP7 G3, G4, G5, G11 and VP4 P[6] and P[7] genotypes were detected in cows [21]. Additionally, bovine–porcine RVA with advantageous genetic configurations retains the ability to infect and cause disease in its heterologous host [34,35]. Furthermore, avian G17P[17] and G18P[17] strains were also reported in cattle [19,36]. Together, it provides evidence for porcine/avian-to-bovine interspecies transmission events [21,37]. Additionally, bovine VP7 G1, G2, G3, G4, G12 and VP4 P[6] genotypes were detected in human RVA, emphasizing the zoonotic capacity of RVA.

Pigs

We found 62 porcine RVA genomic constellations, of which 27 were complete (Table A4), 27 had partial information and 8 were mixed genotypes (Appendix A). The Wa-like genomic constellation G5-P[7]-I5-R1-C1-A1-N1-T1-E1-H1 is the most prevalent in pigs. However, we found two complete genomic constellations with a DS-1 structure, reported in South Africa and Thailand [38].

The VP7-VP4 combinations in pigs were found to be diverse: 47 GP combinations were reported, but none were superior to the original 6%, except for G5P[7] with a distribution of 37.3% [19]. Two other frequent combinations were G5P[6] and G4P[6] (Appendix A) [19].

Regarding the VP6 gene, the I5 genotype is present in most of the genomic constellations (86%), followed by I1 (11%) and I2 (1.8%) (Appendix A).

The RVA G12 genotype is rapidly emerging as an important human pathogen [24]. The first report of an animal RVA strain with G12 genotype specificity found only in one pig, the RVA/Pig-wt/IND/RU172/2002/G12P[7] [21,39]. However, another strain (RVA/Pig-wt/UGA/BUW-14-A008/2014/G12P[8]) with a Wa-like backbone has been described recently in Uganda [33].

Typical swine RVA strains may be able to infect other species through interspecies transmission [27]. In addition, several studies have demonstrated that the reassortment of genomic segments between porcine and bovine RVA strains does occur [19]. For example, porcine-like RVA G5P[7] strains were found in Korean cattle herds and vice versa; moreover, bovine-like RVA G6P[1] strains were sporadically detected in some Argentinean pig herds [40]. Partial genomic analysis of two RVA/Pig-wt/IND/HP113/2002/G6P[13] and RVA/Pig-wt/IND/HP140/2002/G6P[13] porcine strains revealed bovine-like VP6 and VP7 genes, combined with porcine-like NSP4 and NSP5 genes, providing evidence for bovine–porcine interspecies transmission and reassortment events. Another study confirmed the direct transmission of RVA from pigs to wild boars in Japan [41]. Moreover, co-circulation of the same GP genotypes in pigs and wild boar populations was confirmed by phylogeny; these results also suggest that natural reassortment events occurred before or after transmission [41].

Horses

We found 10 complete constellations in horses (Table A5). The typical genomic constellation obtained from several European, American and African RVA strains showed a peculiar composition: G3/G14P[12]-I2/I6-R2-C2-M3-A10-N2-T3-E2/E12-H7 [20]. Furthermore, single reports of unusual genomic constellations were found, such as the porcine-like constellation (G5-P[7]-I5-R1-C1-M1 -A8-N1-T1-E1-H1) and feline/canine (G3-P[3]-I3-R3-C3-M3-A9-N3-T3-E3-H6). We also found two bovine genomic constellations (G6-P[5]-I2-R2-C2-M2-A13-N2-T6-E2-H3 and G8-P[1]-I2-R2-C2-M2-A14-N2-T6-E2-H3) and the single RVA/Horse-tc/GBR/L338/1991/G13P[18] genomic constellation (Table A5). Most of the constellations were reported in Japan (60.9%), followed by Argentina (34.5%) and USA (11%) [20].

The most prevalent VP7-VP4 combinations RVA genotypes were G3P[12] followed by G14P[12], as previously reported [20]. Other combinations reported were G8P[1], G3P[3], G5P[7] and G13P[18] [20]. We also found other single reports: G6P[5], G6P[1], G10P[1] and G10P[11] (Appendix A) [42].

Regarding the VP7 gene, the G3 and G14 genotypes were the most frequent (54.7% and 44.3%, respectively), whereas other genotypes were represented by less than 1% (Appendix A). The same pattern is observed in VP4, P[12] were reported in 95% of the cases and the rest of the genotypes were present in less than 1% (Appendix A).

In our findings, I6 and I2 have the highest number of reports: 54.3% and 43.2%, respectively (Appendix A). The likely ruminant origin of the E12 NSP4 genotype, exclusively carried by the South American equine, bovine, new-world camelids and small ruminant RVA strains, was described elsewhere [43]. Interestingly, E12 was also reported in humans [44].

In concordance with previous reports, equine RVA genotype combinations have not been seen in other host species [20]. Recent findings suggest that several unusual human and equine RVA strains might have originated in bats, based on their phylogenetic clustering, despite various levels of nucleotide sequence identities between them [45].

Since 2013, equine-like G3 RVA strains have been detected in human throughout the world and these strains were found to be dominant in some countries [46]. There were no significant differences in the clinical characteristics of equine G3 and non-equine G3 RVA infections. On the other hand, equine G3 strains with DS-1-like genomic constellations were detected in Japan [46].

Hoofed mammals

Rotaviruses have been reported in many exotic hoofed mammals, such as alpacas, antelopes, bison, boars, deer, camels, gazelles, guanacos, gnus, giraffes, impalas, okapi, reindeer, wild hogs and vicunas [21,23]. For many of these hosts, the viruses were detected by electron microscopy, electro-pherotype or serology and no genotypes were obtained (Table A1).

We found 24 genomic constellations of ungulates, excluding horses, cows and pigs (Table A6). Of these, 13 genomic constellations belong to camelids, which will be analyzed in the camelids section.

Japanese wild boars were reported to carry G9P[23], G4P[23], G9P[13] and G4P[6] genotypes; moreover, pig-to-wild-boar interspecies transmissions were also demonstrated [41].

The giraffe genomic constellation was found to be closely related to bovine RVA strains. The VP7-VP4 reported genotype was G10P[11,26].

Concerning goats, three combinations G8P[1], G6P[14], G10P[14] and G10P[5] were described in many areas worldwide such as Asia [47,48,49], South Africa [16,50] and Argentina [51].

Two buffalo strains were detected in South Africa and its genomic constellations were described. Both strains presented a conserved, typical artiodactyl constellation [52]. Interestingly, the RVA/Buffalo-wt/ZAF/4426/2002/G29P[14] strain carried a VP7 genotype different from other ungulates, the G29 genotype, which was described only three times before: twice in humans, and once in a cow [52]. The RVA RVA/Buffalo-wt/ZAF/4426/2002/G29P[14] strain is the second P[14] genotype strain detected in a South African wildlife species; the first was the antelope RVA/Antelope-wt/ZAF/RC-18/08/2008/G6P[14] strain [52].

The VP7 G8 genotype was reported in all camelids (12 reports), as well as cows (99), goats (13), sheep (2) and deer (1). However, it was also reported in dogs (9), monkeys (3) and humans (91) (Appendix A).

The VP4 P[14] genotype has been detected in combination with one of the three major bovine VP7 genotypes (G6, G8 or G10) in different ungulate host species [21,24]. In addition, human P[14] strains were found to be closely related to ungulates [24]. Therefore, large ruminants (cattle and antelope), small ruminants (goat and sheep) and other ungulate species, such as camelids, may play roles as sources of infectious RVA P[14] strains to humans [24]. The P[14] genotype is also prevalent in rabbits (G3P[14]); however, previous phylogenetic studies have suggested that lapine RVA might have a distant evolutionary relationship with human and artiodactyl P[14] RVA strains [24,53].

Whole genomic analyses of a sable antelope RVA strain and two guanacos RVA strains provided evidence for common origin of these strains with ovine and other ruminants’ RVs, corroborating the presence of an overall consensus genomic constellation G6/G8P[14]-I2-R2/R5-C2-M2-A3/A11-N2-T6-E2/E12-H3 that might be circulating among ruminants, camelids and other artiodactyl host species [21,24]. In sheep, goats, deer, sable antelopes and giraffes, the genomic constellation backbone is similar to that of cows:I2-R2-C2-M2-Ax-N2-T6-E2-H3 [24,47,49,54].

Camelids

RVA were detected in alpacas [29,55,56], guanacos [57], vicunas [58], llamas [59] and camels [60,61]. In total, 14 complete genomic constellations were found, eight of them were partial (Table A6). Moreover, 259 different genotypes were reported in these hosts.

The camel RVA/Camel-wt/KUW/s21/2010/G10P[15] strain contained VP2, NSP2, NSP3 and NSP5 genes with high nucleotide sequence similarities to those of ovine and bovine RVA [60]. Conversely, its VP1 gene shared nucleotide sequence identities with porcine RVA strains. The VP4, VP6 and VP7 genes of the camel RVA strain exhibited nucleotide sequence similarities to those of reference strains from ruminants [60].

As we mentioned above, South American hosts possess a NSP4 E12 genotype [24,43]. However, there is a report of a vicuna carrying the E3 genotype [29]. Moreover, the alpaca RVA genes are related to RVA sequences from cows, guanacos and vicunas [29], and it was also reported to be related to human and pig RVA [55].

The guanaco RVA strains have a close relationship with each other and were found to have close evolutionary relationships with vicuna RVA strains [57]. In addition, the ovine, antelope, and G6P[14] human strains were also related to several gene segments [21,24]. Moreover, guanacos and cattle come in close contact with sheep and goats, which might be the intermediate carriers and/or might be the main hosts of the P[14] strains [24].

Avian

Exclusive and variable RVA genomic constellations were found in avian hosts [21]. The complete genomes of chicken (4), pheasant (1), dove (3), turkey (3), gull (1), crow (1) and velvet scoter (1) RVA strains have been analyzed so far. In addition, the partial genomic constellations of chicken (4), turkey (1), guinea fowl (2) and dove (1) RVA strains have also been reported (Table A7).

The VP7 genotypes reported in avian hosts were G7, G17, G18, G19, G22 and G23. The G7 genotype was reported in chickens and turkeys, with G17 only in doves, and G19 had 75 reports in chickens and 18 in guinea fowls. G18 was reported in chickens (1), doves (24) and velvet scoters (1). Finally, G22 was reported in turkeys (Appendix A).

Regarding the VP4 gene, the reported genotypes were P[17], P[30], P[31], P[35] and P[37] (Appendix A). There are VP6 genotypes exclusively found in birds such as I11 (69 chicken, 9 guinea fowl, 1 dove) and I21 (1 common gull, velvet scoter and 1 dove) genotypes (Appendix A).

Avian VP7-VP4 genotype combinations are similar in doves, velvet scoters (duck) and guinea fowl. Two species reported in Japan (RVA/JungleCrow-wt/JPN/JC-105/2019/G40P[56] and racoon RVA/Racoon-tc/JPN/Rac-311/2011/G34P[17]) have similarities in their genomic constellation’s backbone except for VP7 and VP4 genotypes [21]. Interestingly, velvet scoters and gulls, also reported in Japan, have very different genomic constellations (Table A7). It is out of the aim of this study to understand the evolutionary history of RVA and its hosts, but these data could be important to address in further investigations. 

Based on phylogenetic analyses, the avian RVA strains were found to be closer to each other than non-avian strains. Moreover, they were genetically divergent from mammalian hosts, and there were few differences between them [62]. Moreover, most of the chicken RVs lack the putative open reading frame (ORF) encoding the NSP6 protein, which is retained in gene segment 11 of most mammalian RVA [63].

There is evidence for reassortment events and insertions/deletions in the untranslated regions (UTR) of the gene segments of avian RVA strains [21]. Additionally, interspecies transmissions were reported for avian RVA. Briefly, the bovine G6P[1] strain was reported in turkeys and the turkey NSP4 E2 genotype clustered together with bovine RVA [64]. The avian VP7 G18 genotype was found in cows [65] and red foxes [66]. Moreover, the VP4 P[17] genotype was also reported in cows and racoons, and the P[35] genotype was reported once in cows [21,36,65]. In addition, the red fox genomic constellation is closely related to avian genotypes and possesses the genetic backbone of the avian strain PO-13 [66].

Canine/feline RVA

We found the genomic constellations of 4 dogs and 4 cats (Table A8). The typical backbone for cats and dogs is G3-P[3]-I3-R3-C2-M3-A9-N2-T3-E3-H6 [28]. Rotavirus strains bearing a G3P[3] genotype are typically found in dogs, cats, bats, monkeys, rats and mice [67,68]. The published data from the USA, Japan, Italy and South Korea indicate that all of the canine RVA strains bear G3P[3], whereas feline RVA shows G3P[3] and G3P[9] [67].

The feline RVA/Cat-wt/ITA/BA222/2005/G3P[9] strain was found to possess several genes related to artiodactyl RVA, and a human RVA-like NSP2 gene belonging to the N1 genotype [57,69]. However, several preliminary studies have reported strains which could belong to the BA222-05-like genomic constellation , suggesting that this strain might be circulating in both feline and canine/feline-like human RVA worldwide [57]. Another feline strain, RVA/Cat-tc/AUS/Cat2/1984/G3P[9], was speculated to be derived from multiple reassortment events involving canine, feline, human and bovine RVA [67]. Human G3 strains with novel P[3] or P[9] genotypes appear to have originated from direct transmission from cats or dogs to humans, as some G3P[3] or G3P[9] strains detected in humans are virtually identical to cat or dog strains of the same genotype [28].

Moreover, the canine/feline genomic constellation was identified in bats (RVA/Bat-wt/BGR/BB89-15/2008/G3P[3]) [70] and horses (RVA/Horse-wt/ARG/E3198/2008/G3P[3]) [20], and four types of AU-1-like genotype constellations were identified in rodents (strains RVA/Rat-wt/CHN/LQ6/2013/G3P[3], RVA/Rat-wt/CHN/LQ285/2013/G3P[3], RVA/Rat-wt/CHN/RA108/2013/G3P[3] and RVA/Rat-wt/CHN/RA116/2013/G3P[45]) [71]. The diverse host range of AU-1-like genomic constellations appears to be related to multiple reassortment events involving bats, feline species and rodents [71] (Table A8).

The dog RVA/Dog-wt/GER/88977/2013/G8P[1] strain has been reported as having a genomic constellation characteristic of G8P[1] bovine-like RVA [68,72]. Sieg and colleagues (2015) suggested that the canine RVA/Dog-wt/GER/88977/2013/G8P[1] strain might have evolved from a human RVA through multiple reassortment events with artiodactyl (possibly bovine) RVA strains. This strain has been reported to cross barrier species and also infect humans [68].

Non-human primates

To date, RVA has been reported in many simian species (Table A1). Seven simian genomic constellations have been reported: RVA/Simian-tc/USA/RRV/1975/G3P[3], RVA/Rhesus-wt/USA/TUCH/2002/G3P[24], RVA/Simian-tc/ZAF/SA11-Both/1958/G3 P[2], RVA/Rhesus-wt/USA/PTRV/1990/G8P[1], RVA/Simian-tc/ZAF/SA11-H96/1958/G3P[2], RVA/Simian-tc/ZAF/SA11-5S/1958/G3P[1] and, RVA/Simian-wt/KNA/08979/2015/G5P[X] (Table A9). Among them, the RVA/Simian-tc/ZAF/SA11-H96/1958/G3P[2] and RVA/Simian-tc/USA/RRV/1975/G3P[3] strains have been used extensively as standard models to study the replication and pathogenesis of rotaviruses [21]. The different laboratory derivatives of SA11 (SA11-5S and SA11-Both) were found to possess bovine RVA genes on a SA11 genetic backbone, possibly resulting from tissue culture contamination of SA11 with the ‘‘O’’ agent, a bovine G8P6[1] RVA strain [73].

Five VP7-VP4 combinations were found for simian strains, because the RVA/Simian-wt/KNA/08979/2015/G5 P[X] RVA/Simian-wt/KNA/08979/2015/G5P[X] strain lacked any VP4 information. The combinations reported were G3P[1], G3P[2], G3P[3], G3P[24] and G8P[1] [73,74,75].

The RVA/Simian-tc/ZAF/SA11-H96/1958/G3P[2] strain was found to be genetically distant to any other known RVs, except for a human strain B10 from Kenya, which was shown to be a zoonotic inter-specie transmission with possible simian origin [74]. Most gene segments of the RVA/Simian-tc/USA/RRV/1975/G3P[3] and RVA/Rhesus-wt/USA/TUCH/2002/G3P[24] strains exhibited genotypes more typical of canine/feline RVA, as mentioned above. However, within these genotypes, low levels of genetic relatedness were observed between RVA/Simian-tc/USA/RRV/1975/G3P[3] and RVA/Rhesus-wt/USA/TUCH/2002/G3P[24] strains and the typical canine/feline or canine/feline-like RVA, indicating that any possible interspecies transmission of a canine/feline progenitor strain to simian host species has not recently happened [74]. Since RVA/Rhesus-wt/USA/PTRV/1990/G8P[1] genes have genotypes and encode outer-capsid proteins similar to those present in bovine RVA, it seems probable that this strain has been transmitted from a bovine or other ruminant and introduced into pig-tailed macaques (where it was reported) [74].

Rodents

Three complete and two partial mouse RVA genomic constellations and five complete rat RVA genomic constellations were found (Table A10) [71,76,77]. RVA has also been detected in squirrels by serological studies and electron microscopy [78,79]; however, the genotype was not determined.

Rodentia is the largest order of mammals distributed globally and is also the largest zoonotic source of human infectious diseases [71]. However, to date relatively little attention has been directed toward the RVA that might circulate in rodent populations [71,77,80]. Given that rats often live high densities nearby to humans and domestic animals, they might play an important role in the cross-species transmission of RVA to these populations [71]. Previous studies demonstrated cross-species transmission between pigs and rats on a pig farm in Brazil, and between rats and humans in China [71,81].

Bats

To date, 23 bat RVA genomic constellations have been reported: 18 complete and 5 partials (Table A11). Our results suggest that the genomic constellations of bats have no common structure, and samples from the same country show a diverse genotype composition. Interestingly, bats do not have a Wa-, DS-1- or AU-1-like structure.

There are 11 different VP7-VP4 genotype combinations reported in bats, of which only G3P[2] and G3P[3] are not exclusive to bats (Appendix A). Bats have a mix of genotypes, including several genotypes reported only in them, such as G25 (3), G30 (4), G31 (1), G33 (1), G36 (2) and G38 (1). Moreover, G20 was reported in bats (2) and humans (2) [45,82]. VP4 has five genotypes reported exclusively in bats, P[43], P[44], P[47], P[48], P[51], P[53] and P[54]. Additionally, P[42] has been reported in bats (3) and humans (1) [83].

Recently, bats have gained relevance and many recent studies have reported new genotypes and prevalence in different populations; in particular, a new RVA species (Rotavirus J) was reported in this host [45,84]. A recent study investigated samples from a variety of bat families and countries: Bulgaria, Romania, Germany, Gabon, Ghana, Costa Rica, Zambia, Cameroon, Kenya, Saudi Arabia, China, France, Brazil and Zambia (Table A1) [45].

The RVA/Bat-tc/MSLH14/2012/G3P[3] strain was successfully isolated from lesser horseshoe bats in China. Its genomic constellation is similar to feline/canine-like human strains. Phylogenetically, the RVA/Bat-tc/MSLH14/2012/G3P[3] strain appears to be distant from the feline/canine RVA strain, and it is believed to represent a true bat RVA strain [70].

Order Carnivora

RVs have been reported in several other carnivores such as racoons, badgers, coyotes, civets, red foxes, skunks, wild dogs and bears (Table A1 and Table A12). Nevertheless, the detection of RVA in most of these hosts was carried out by serology or electron microscopy, and the RVA sequence strains were not yet determined (Table A1). Interestingly, many of the reports in all of these carnivore animals seem to have a possible interspecies transmission event. It was already mentioned that the RVA strain obtained from red foxes was related to the avian RVA/Pigeon-tc/JPN/PO-13/1983/G18P[17] strain [66]. The VP7 gene of the giant panda RVA/Giant panda-tc/CHN/CH-1/2008/G1P[7] strain was found to be closely related (nucleotide sequence identity of 98.5%) to porcine G1P[7] RVA/Pig-wt/CHN/sh0902/2008/G1P[7] strain [85]. Conversely, the VP4 and NSP4 genes were related to porcine-like RVA strains [85]. Additionally, the VP7 and VP4 genes described in racoon dogs and masked palm civets in Japan were closely related to each other [27]. The authors highlight that G3P[9] is widely spread in wild animals, and that these wild animals could have been infected by RVA G3P[9] through interspecies transmission from humans or cats [27].

Other animals

A RVA strain from shrews has been identified recently in China due to its possible role as a reservoir and its proximity with human populations [71]. The genomic constellation described in shrews has a close relationship with another mouse strain found in the same study (Table A13) [71].

RVA strains have been detected in rabbits and hares in different countries; most of these were found to carry the G3P[14] or G3P[22] genotype [86,87]. In addition, a G6P[11] strain was also reported in these hosts [88].

There are only three reports on detection of RVA in marsupial species: a kangaroo [23], a common opossum [89], and sugar glider [90], but only the last species has been characterized genetically. It is thought that RVA strains in sugar gliders have evolved uniquely, possibly not only in sugar gliders but also in other marsupial species [90].

There were also reports of RVA found in aquatic mammals (such as seals or sea lion) and fishes in nature; however, there is no conclusive information on disease caused by RVA [91,92,93]. In a study with young sea lions, clinically healthy individuals tested positive for RVA through serological methods [92]. Shellfish, which are an important source of food for humans, can concentrate these viruses due to their filter feeding process [93]. This scenario raises a public health threat as shellfish are often consumed raw or improperly cooked [94]. Recently, infectious RVA was recovered from shellfish in Argentina, highlighting the relevance to analyze it as a food-borne virus [30]. Interestingly, none of these genetic groups share close sequence homology with an SA11 RVA [91].

Vaccines

The WHO recommends that rotavirus vaccines should be included in all national immunization programmes [95]. To date, six commercial vaccines have been licensed for humans: (i) Rotarix^®^ (GSK) a monovalent vaccine composed of a live attenuated RVA human RIX4414 G1P1A [8] strain; (ii) RotaTeq^®^ (Merk), a pentavalent vaccine composed of five RVA/Cow-tc/USA/WC3/1981/G6P[5] reassortant bovine strains, which express VP7 and VP4 outer capsid proteins and G1, G2, G3, G4 and P[8] genotypes, respectively; (iii) Rotavac^®^ (Bharat) containing an attenuated human 116E (G9P[11]) strain; and (iv) RotaSiil^®^ (Serum Institute) containing 5 single gene (VP7) substitution reassortments between the human strains G1, G2, G3, G4, G9 and the bovine UK G6P[5] strain. This vaccines are considered highly effective in preventing severe gastrointestinal disease by the WHO and are available internationally [95]. (v) Rotavin-M1 (Polyvac, Vietnam) is composed of a monovalent rotavirus RVA/Human-wt/VN/KH0118/2003/G1P[8] strain; and (vi) Lanzhou lamb rotavirus (Lanzhou Institute of Biological Products, China) is composed of a lamb G10P[12] strain [96].These last two vaccines are available in the countries where they are produced, but are not available internationally [95].

For the prevention of neonatal calf diarrhea, although vaccination is not performed routinely [19], the currently available vaccines are: Guardian1^®^ (Merk), which contains a G6P[1] and G6P[5] strains, Scourguard 3^®^ (Pfizer) composed of RVA/Cow-tc/USA/NCDV-Lincoln/967/G6P[1], Scourguard 4KC^®^ (Pfizer) composed of serotypes G6 and G10, and Trivaction 6^®^ (Merial) [19]. The ProSystems ROTA^®^ (Merck) is a polyvalent live oral RVA vaccine which contains strains RVA/Pig-tc/USA/OSU/1975/G5P[7] and RVA/Pig-tc/USA/Gottfried/1975/G4P[6] and RVA/Pig-tc/USA/A2/19XX/G9P[7] strains. In horses, inactivated vaccines have been developed to prevent diarrhea caused by RVA [20]. The USA, United Kingdom and Ireland currently use the Fort Dodge Animal Health and the Zoetis vaccines, which contain an RVA/Horse-wt/GBR/H-2/1976/G3P[12] strain. In Argentina, the ROTAMIX EQUIN^®^ (Biochemiq) is a polyvalent vaccine composed of RVA RVA/Horse-wt/GBR/H-2/1976/G3P[12], RVA/Simian-tc/ZAF/SA11-H96/1958/G3P[2] and RVA/Cow-tc/USA/NCDV-Lincoln/967/G6P[1] strains. In Japan, the inactivated vaccine HRV (Nisseiken), which contains RVA/Horse-tc/JPN/HO-5/1982/G3BP[12], is currently available [20].

To date, no vaccines have been developed for avian RVA [25,26].

## 5. Conclusions

Studies of rotaviruses from livestock, horses, companion animals and other species are limited, and the genomes of only a few of these strains have been analyzed so far. Both the reassortment and the interspecies transmission of RVA strains contribute to generate diversity among the RVA population and could become an increasingly dangerous pathogen affecting human health. We observed that reassortment occurs frequently, and many genotypes were combined and tested constantly. These variants easily pass to other species with unpredictable results. This raises the question as to when a new virulent variant will appear, and what impact it will have. Continuous surveillance for the distribution of RVA strains around the animal world is critical to prevent outbreaks, and for monitoring the circulating rotavirus strains for the continued development of new rotavirus vaccines and for evaluating vaccine impact. In addition, knowing which genotypes affect a particular species allows for the design of vaccines with a wide coverage of genotypes but aimed at that particular species (or group). Moreover, knowing the reservoir host of a virus, it is possible to assess the risk of emergence and prevent future outbreaks. Further evolutionary studies are needed to understand how its broad RVA spectrum affects the reassortment, besides its impact on the fitness of the virus.

## Data Availability

Not applicable.

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
