# Peer review of "Zoonotic RVA: State of the Art and Distribution in the Animal World"

_viruses, 2022, doi:10.3390/v14112554_

Round 1
Reviewer 1 Report
The current review article of group A rotavirus (RVA) strains of animal origin represents a major undertaking by the authors and will provide the rotavirus field with a much-needed review of the genotype constellations identified in several animals species on a global basis. With the advent and ease of sequencing, genotyping of rotavirus strains has provided a closer look at the molecular evolution and relationships among different animal strains and human strains. The review focuses on the vast diversity of animal strains and summarizes, in a systematic way, the different genotype constellations identified to data among RVA animal strains.
The review article would benefit from the following modifications:
1) The authors should include in the introduction (section 1 in article) that there are now ten (A to J) rotavirus species identified to date, with RVA being the most common and widely studied due to its prevalence worldwide in both mammals (including humans) and birds. Likewise, the authors should acknowledge that besides the ratified rotavirus species, there are currently two additional rotavirus species (K and L) being considered. A good point of reference could be the recent publication by Reimar et al., 2022. Once these facts have been acknowledged, the authors can state that the current review article will focus on RVA animal strains.
2) Additionally, the characterization and nomenclature of the rotavirus genome that should be included in the introduction section because these are the basis of the classification of rotavirus strains.
3) Likewise, it is important to state up front that gene reassortment is one of the main drivers of rotavirus diversity.
4) The authors should add a section detailing the current approved rotavirus vaccines for humans as well as the different rotavirus vaccines for domestic animals (pigs, cattle, horses). Inclusion of this information, along with the description of the components of the vaccines, will add value to the review article and allow the reader to gain further understanding of the rotavirus diversity and importance of implementation of vaccine strategies in the animal industry.
5) The review article would benefit from a careful review of the English grammar. For example:
Line 12: Replace “lower than” with “under”
Line 31: replace “young animals and birds species” with “young mammals and birds”
Line 77: not sure what it is meant by “At the beginning”. Please specify actual time frame. Or reword sentence.
Author Response
Dear Reviewer 1
Thank you for helping strengthened our manuscript.
The current review article of group A rotavirus (RVA) strains of animal origin represents a major undertaking by the authors and will provide the rotavirus field with a much-needed review of the genotype constellations identified in several animals species on a global basis. With the advent and ease of sequencing, genotyping of rotavirus strains has provided a closer look at the molecular evolution and relationships among different animal strains and human strains. The review focuses on the vast diversity of animal strains and summarizes, in a systematic way, the different genotype constellations identified to data among RVA animal strains.
The review article would benefit from the following modifications:
- The authors should include in the introduction (section 1 in article) that there are now ten (A to J) rotavirus species identified to date, with RVA being the most common and widely studied due to its prevalence worldwide in both mammals (including humans) and birds. Likewise, the authors should acknowledge that besides the ratified rotavirus species, there are currently two additional rotavirus species (K and L) being considered. A good point of reference could be the recent publication by Reimar et al., 2022. Once these facts have been acknowledged, the authors can state that the current review article will focus on RVA animal strains.
As request the reviewer, information about the Rotavirus species classification was included (line 50 - 56).
“Different rotavirus species are all grouped into the genus Rotavirus, including 9 species (or groups) designated as RVA to RVD and RVF to RVJ, officially recognized by the International Committee on Taxonomy of Viruses (ICTV) [7]. Other proposed species designated as RVK and RVL [8], are considered preliminary due to partial genome sequences available so far [7]. In this work we focused on RVA (Rotavirus species A), which is the most common and widely studied viral group because of its high prevalence worldwide in both mammals (including humans) and birds.”
- Additionally, the characterization and nomenclature of the rotavirus genome that should be included in the introduction section because these are the basis of the classification of rotavirus strains.
As request the reviewer, information about the characterization and nomenclature of the rotavirus was included (line 37 - 49).
“RVA has a segmented double-stranded RNA genome which consists of 11 segments enclosed in a triple-layered icosahedral capsid, encoding six viral structural proteins (VP1 to VP4, VP6, and VP7) and six nonstructural proteins (NSP1 to NSP6) [5]. Each genome segment, with the exception of gene 11 that encodes two proteins (NSP5 and NSP6), are monocistronic. The inner layer of the rotavirus virion is mainly composed of VP2, which encases VP1, the viral RNA-dependent RNA polymerase, and VP3, the viral capping enzyme [5]. The middle layer of the virion are VP6 trimers; VP7 and spikes of VP4 compose the outer layer. All nonstructural proteins with the exception of NSP4, interact with nucleic acids and are involved in viral replication processes; NSP4 is responsible for morphogenesis [5].
In this review it was considered the nomenclature based in the notation Gx-P[x]-Ix-Rx-Cx-Mx-Ax-Nx-Tx-Ex-Hx employed for the VP7-VP4-VP6-VP1-VP2-VP3-NSP1-NSP2-NSP3-NSP4-NSP5 encoding genes, that compares complete rotavirus genomes[6].”
- Likewise, it is important to state up front that gene reassortment is one of the main drivers of rotavirus diversity.
As request the reviewer, information about rotavirus gene variation was included (line 57 - 59).
“The main drivers of rotavirus diversity appear to be point mutations that occur con-tinuously based on the high error rate of the RV polymerase, and genome reassortments occurring by strain coinfections in hosts, often involving zoonotic transmission [1].”
- The authors should add a section detailing the current approved rotavirus vaccines for humans as well as the different rotavirus vaccines for domestic animals (pigs, cattle, horses). Inclusion of this information, along with the description of the components of the vaccines, will add value to the review article and allow the reader to gain further understanding of the rotavirus diversity and importance of implementation of vaccine strategies in the animal industry.
As request the reviewer, a new section about vaccines for humans and animals was included (line 500 - 530).
“The WHO recommends that rotavirus vaccines should be included in all national immunization programmes [150]. To date, six commercial vaccines have been licensed for humans: i) Rotarix® (GSK) a monovalent vaccine composed by a live attenuated RVA human RIX4414 G1P1A[8] strain; ii) RotaTeq® (Merk), a pentavalent vaccine composed by five RVA/Cow-tc/USA/WC3/1981/G6P[5] reassortant bovine strains, which express VP7 and VP4 outer capsid proteins G1, G2, G3, G4 and P[8] genotype respectively; iii) Rota-vac® (Bharat) that contains an attenuated human 116E (G9P[11]) strain and; iv) RotaSiil® (Serum Institute) containing 5 single gene (VP7) substitution reassortants between human strains G1, G2, G3, G4, G9 and the bovine UK G6P[5] strain. This vaccines are considered highly effective in preventing severe gastrointestinal disease by the WHO and are availa-ble internationally [150]. v) Rotavin-M1 (Polyvac, Vietnam) is composed by a monovalent rotavirus RVA/Human-wt/VN/KH0118/2003/G1P[8] strain; and vi) Lanzhou lamb rota-virus (Lanzhou Institute of Biological Products, China) is composed by a lamb G10P[12] strain [151].This last two are available in the countries where they are produced, but are not available internationally [150].
For prevention of neonatal calf diarrhea, although vaccination is not performed rou-tinely [19]. The currently available vaccines are: Guardian1® (Merk) that contains a G6P[1] and G6P[5] strains, Scourguard 3® (Pfizer) composed by RVA/Cow-tc/USA/NCDV-Lincoln/967/G6P[1], Scourguard 4KC® (Pfizer) composed by serotypes G6 and G10 and, Trivaction 6® (Merial) [19]. The ProSystems ROTA® (Merck) is a polivalent live oral RVA vaccine which contains strains RVA/Pig-tc/USA/OSU/1975/G5P7 and RVA/Pig-tc/USA/Gottfried/1975/G4P[6] and RVA/Pig-tc/USA/A2/19XX/G9P[7] strains. In horses, inactivated vaccines have been de-veloped to prevent diarrhea caused by RVA [20]. The USA, United Kingdom and Ireland currently used the Fort Dodge Animal Health and the Zoetis vaccines which contains an RVA/Horse-wt/GBR/H-2/1976/G3P[12] strain. In Argentina the ROTAMIX EQUIN® (Bio-chemiq) is a polyvalent vaccine composed by RVA RVA/Horse-wt/GBR/H-2/1976/G3P[12], RVA/Simian-tc/ZAF/SA11-H96/1958/G3P[2] and RVA/Cow-tc/USA/NCDV-Lincoln/967/G6P[1] strains. In Japan, the inactivated vaccine HRV (Nisseiken) which contains RVA/Horse-tc/JPN/HO-5/1982/G3BP[12] is currently available[20].
To date, no vaccines have been developed for avian RVA [75,76].”
- The review article would benefit from a careful review of the English grammar. For example:
Line 12: Replace “lower than” with “under”
As requested the reviewer, the term was replace.
Line 31: replace “young animals and birds species” with “young mammals and birds”
As requested by the reviewer, the sentence was corrected.
Line 77: not sure what it is meant by “At the beginning”. Please specify actual time frame. Or reword sentence.
As requested by the reviewer, the sentence was reword.

Reviewer 2 Report
This review article is well written and contains a piece of very good information on human monkeypox. I am recommending some minor revisions before its publication as mentioned below.
1. In the introduction change the values (122,000 to 216,000 deaths) to the nearest million.
2. In the materials and methods, provide the correct link for the BLAST.
3. In Table A.1., correct the spelling of chicken in the group avian.
4. In the subsection "Common GP[] genotype combinations", what is the meaning of [], clarify it.
5. There are a few grammatical mistakes in the text. I request authors check the manuscript before submission of the revised version.
6. Some of the references are not as per the format of the journal. Update it accordingly.
Rest is ok.
Congratulations to all the authors for the good article.
Author Response
Dear Reviewer 2
Thank you for helping strengthened our manuscript.
This review article is well written and contains a piece of very good information on human monkeypox. I am recommending some minor revisions before its publication as mentioned below.
- In the introduction change the values (122,000 to 216,000 deaths) to the nearest million.
As requested the reviewer, the sentence was corrected (line 33 - 35).
“Annually, RVA are responsible for almost 500,000 deaths in under-5-year-old infants, mainly in developing countries [2,3].”
- In the materials and methods, provide the correct link for the BLAST.
As requested the reviewer, the link was corrected (line 123).
(https://blast.ncbi.nlm.nih.gov/Blast.cgi)
- In Table A.1., correct the spelling of chicken in the group avian.
As requested the reviewer, the spelling was corrected.
- In the subsection "Common GP[] genotype combinations", what is the meaning of [], clarify it.
As requested the reviewer, the term was clarify (line 156).
“There are common combinations of VP7 and VP4 protein (GP[x]) as G8P[1]…”
- There are a few grammatical mistakes in the text. I request authors check the manuscript before submission of the revised version.
As requested the reviewer, the grammar was checked.
- Some of the references are not as per the format of the journal. Update it accordingly.
Rest is ok.
As requested the reviewer, the bibliography was update.
